# Is It Possible to Personalize the Diagnosis and Treatment of Breast Cancer during Pregnancy?

**DOI:** 10.3390/jpm11010018

**Published:** 2020-12-28

**Authors:** Petra Tesarova, David Pavlista, Antonin Parizek

**Affiliations:** 1Department of Oncology, 1st Faculty of Medicine Charles University and General University Hospital, 128 08 Prague, Czech Republic; 2Department of Gynecology and Obstetrics, 1st Faculty of Medicine Charles University and General University Hospital, 128 08 Prague, Czech Republic; david.pavlista@vfn.cz (D.P.); antonin.parizek@vfn.cz (A.P.)

**Keywords:** breast cancer, pregnancy, chemotherapy, tailoring, personalization

## Abstract

The main goal of precision medicine in patients with breast cancer is to tailor the treatment according to the particular genetic makeup and the genetic changes in the cancer cells. Breast cancer occurring during pregnancy (BCP) is a complex and difficult clinical problem. Although it is not very common, both maternal and fetal outcome must be always considered when planning treatment. Pregnancy represents a significant barrier to the implementation of personalized treatment for breast cancer. Tailoring therapy mainly takes into account the stage of pregnancy, the subtype of cancer, the stage of cancer, and the patient’s preference. Results of the treatment of breast cancer in pregnancy are as yet not very satisfactory because of often delayed diagnosis, and it usually has an unfavorable outcome. Treatment of patients with pregnancy-associated breast cancer should be centralized. Centralization may result in increased experience in diagnosis and treatment and accumulated data may help us to optimize the treatment approaches, modify general treatment recommendations, and improve the survival and quality of life of the patients.

## 1. Introduction

The need to protect the fetus from the adverse events associated with the treatment of cancer represents a significant barrier to the implementation of genomic and molecular biological personalization of treatment in a subgroup of pregnant patients with breast cancer. Pregnancy-associated breast cancer (PABC) is defined as breast cancer diagnosed during pregnancy (BCP) or in the first postpartum year or at any time during lactation. BCP is a special situation of concomitant pregnancy and cancer and, due to different subtypes of breast cancer, tumor detection at different stages and diagnosis confirmed at different trimesters of pregnancy does not allow the application of only one standard treatment approach. The reason is also the fact that despite the increasing experience with the treatment of such patients, the published data on PABC are still limited. Prospective studies of breast cancer during pregnancy are almost lacking, and we must rely on data from retrospective case series [1,2]. 

The development of personalized precision medicine as the ultimate aim of the treatment of PABC is dependent on a better understanding of the pathogenesis of PABC [3].

The advent of big genomic data has shifted our attention from examining single genes to whole exome and transcriptome analysis with the aim of identifying new predictive factors, biomarkers, and therapeutic targets although until now, still only some more frequently mutated genes are tested to achieve better cost-effectiveness, i.e., genes that seem to be associated with better cost-effectiveness, enhanced data analysis, and rapid availability for the immediate clinical decisions [4]. Unfortunately, pregnant patients with breast cancer do not yet benefit from these advances in precision medicine.

Tailoring the treatment of breast cancer in pregnancy must primarily adapt to the course of pregnancy. Due to the young age, the disease is more often associated with hereditary mutations of risk genes. Cancer is more likely to have a high histological risk profile and is diagnosed at a more advanced stage. Therefore, in clinical practice we are more often faced with the need to treat patients with a very advanced stage of cancer, frequently with the presence of metastases in the skeleton or visceral organs.

## 2. Epidemiology

Increasing incidence of PABC is associated with an overall increase of breast cancer in the population and increasing age at conception. PABC is still, however, relatively uncommon (with an incidence of 15 to 35 per 100,000 deliveries, more frequently occurring during the first postpartum year rather than during the pregnancy) although breast cancer is the most common type of cancer in pregnancy [5]. PABC is very rare (one per 1000 pregnancies annually, i.e., 0.07% to 0.1% of all malignant tumors, only) [6].

Pregnancy generally has a lifelong protective effect on breast cancer risk, but it increases the risk of breast cancer for several years after pregnancy with the highest risk at 6 years after delivery and significantly higher risk in older primiparas. There are important differences (in terms of diagnosis, treatment, and outcome) between PABC and breast cancer after pregnancy [2].

## 3. Pathophysiology

The pathogenesis of PABC is not fully understood [7]. Pregnancy and lactation are associated with increased levels of estrogens with the impairment of their normal cyclical pattern resulting in resultant molecular and histological changes in the breast gland. Increased estrogen levels may also promote the formation of metastases. Other factors, e.g., immune changes and inflammation [8], also promote carcinogenesis, especially in women with occult disease at conception (more frequent during the involution of the mammary gland) [9]. It should also be stressed that late diagnosis of breast cancer in pregnancy may also contribute to the more frequent presence of metastatic disease.

Pregnancy-associated plasma protein A (PAPP-A) may also play an important role in the development of metastatic PABC (by its collagen-modifying properties) and may help to identify patients at risk of metastatic disease [10].

## 4. Pathology

As in non-pregnant women, the most common form of PABC is infiltrating ductal adenocarcinoma. PABC is, however, less differentiated and (as already stressed) diagnosed at more advanced stage. Inflammatory breast cancer is also more frequent in pregnancy than in non-pregnant women [11]. The molecular pattern of PABC is different, namely in terms of more frequent mutations of the mucin gene family, mismatch repair deficiencies, and other non-silent mutations [12].

Estrogen and progesterone receptor expression seems to be decreased in PABC compared to that in non-pregnant patients with breast cancer (25% vs. 55% to 60%) [13] probably with no significant difference in overexpression of human epidermal growth factor receptor 2 (HER2) ([14,15], Table 1). Despite many differentially expressed genes, there seems to be no correlation between genetic changes and histopathological and clinical characteristics of BCP. Further studies in search of putative novel biomarkers that could identify the subpopulation of women in childbearing age at risk of PABC are warranted [16].

## 5. Precision Medicine in Breast Cancer

Precision medicine involves the identification of molecular signature, biomarkers, and clinical phenotype and the evaluation of their impact in combination with lifestyle and environmental factors on the prevention and treatment of the disease [17]. Cancer biomarkers may be diagnostic, prognostic, predictive, or used to monitor treatment responses. Prognostic biomarkers provide information about a patient’s overall cancer outcome, irrespective of therapy [18]. They can identify high-risk patients who may benefit from more aggressive treatment but provide no information on which patients will most likely derive a clinical benefit from any specific therapy. Conversely, modifiable predictive markers responding to the treatment can indicate the probability of a patient gaining a therapeutic benefit from a specific treatment [19].

Breast cancer can be classified based on gene expression and histology including the expression of estrogen receptor (ER), progesterone receptor (PgR), and human epidermal growth factor receptor 2 (HER2) into several subtypes, characterized as luminal, normal-like, HER2-overexpressing, and triple negative breast cancer (TNBC) [20]. Gene expression profiling is more in-depth and provides more detailed stratification of breast cancer compared to histology itself. Based on these analyses, breast cancer was shown to be very heterogeneous with substantial variability in biological behavior, pathogenesis, response to treatment, and outcome [21].

Analysis based on microarray gene expression is already available, but its cost prevents its broader use in routine clinical practice with more focused analysis aimed at smaller gene sets (breast cancer index, Endopredict, the Oncotype DX 21-gene recurrence score, the BreastOncPx 14-gene distant metastasis signature, 50-gene signature called PAM50 (Prosigna), and the MammaPrint 70-gene prognosis signature) used for breast cancer stratification may emerge as more cost-effective and help clinicians to pinpoint the use of endocrine treatment and adjuvant chemotherapy [22].

To overcome the need to obtain biopsy samples from primary or metastatic lesions, great attention is paid to the blood-based biomarkers, e.g., circulating tumor cells (CTCs), exosomes and circulating tumor DNA (ctDNA), sometimes called liquid biopsy. CTCs are released from the primary tumor and are related to the propensity of the cancer to form distant metastases [23].

Genomic instability, which is common in cancer, results in genetic and epigenetic heterogeneity, and so the outcomes of patients with the same histologic type of cancer may be different in terms of response to treatment and outcome [24].

Epigenetic modification, e.g., DNA methylation and histone acetylation, is instrumental in the early phase of carcinogenesis. Recently, the role of different types of non-coding RNAs (ncRNAs) regulating gene expression and working as epigenetic modifiers has been uncovered [25].

Evaluation of the expression of both estrogen (ER) and progesterone (PR) receptors is indispensable before the introduction of hormonal treatment, and similarly, evaluation of HER2 amplification is necessary for the prediction of the response to anti-HER2 treatment. Mutation of the gene for the estrogen receptor (ESR1) predicts the risk of resistance to aromatase inhibitors. Similar markers predicting the response to radiotherapy and different modes of chemotherapy are warranted [26].

Analysis of some of these biomarkers in clinical practice may refine the search for suitable clinical trials with drugs aimed at the identified targets, but pregnant patients, unfortunately, cannot be recruited to the clinical trials. In the treatment of pregnant women, we can use neither standard, breast-cancer-specific immunohistochemical targets, such as hormone receptor or HER2 antigen positivity, nor targets derived from genomic analysis, such as PIK3 (phosphatidylkinase 3) or ESR1 (gene for estrogen receptor 1) mutations, nor those found by pathologists (TILs (tumor infiltrating lymphocytes)). Off-label treatment aimed at molecular targets not typical for breast cancer (KRas, BRAF, EGFR, etc.) cannot be used in the treatment of PABC.

Pregnancy and concomitantly diagnosed breast cancer are currently a major barrier to the use of precision medicine in the treatment of breast cancer. Its inclusion in treatment plans must be postponed until after delivery or modified so that the questions we specifically address in these situations can be answered. Due to the small number of patients and the fetuses, there are currently no (and will hardly be any in the future) clinical studies in this breast cancer subpopulation.

## 6. Clinical Presentation

Common signs and symptoms of cancer (lump, thickening, change in the size, shape, inverted nipple, etc.) may be hidden because of the pregnancy-associated physiological changes of the breast gland. This can delay diagnosis and adequate care. Patients with the presence of metastases may develop general symptoms, fatigue, back pain, dyspnea, pain and pressure in the right ribs, etc.

## 7. Diagnosis

Physical examination of the breast gland in pregnancy in search for putative cancer is difficult because of pregnancy-associated changes of the breast gland and also the utility of mammography may be limited resulting often in delayed diagnosis of PABC [27]. Any persisting (for more than two weeks) mass should be examined although 80% of the findings in breast biopsies in pregnant women are benign [28]. Mammography is not contraindicated in pregnancy with abdominal shielding (although the decrease of fetal radiation exposure with shielding remains uncertain). The sensitivity of mammography may be decreased due to higher density of the breast gland during pregnancy and lactation, but it still remains useful as a diagnostic tool. Breast ultrasonography can determine whether a breast mass is a simple or complex cyst or a solid tumor without the risk of fetal radiation exposure and may be used to guide the diagnostic biopsy. Gadolinium-enhanced MRI should be (if possible) avoided during pregnancy [29]. Needle core biopsy is the preferred method in any clinically suspicious breast mass and can be safely done during pregnancy, preferably under local anesthesia [30]. Possible infiltration of the lymph nodes by cancer cells should be further evaluated with ultrasound and fine needle aspiration biopsy for cytologic confirmation [31].

## 8. Staging

Modifications of the standard staging work-up should be implemented to protect the fetus (Table 2). Chest radiographs to evaluate for lung metastases should be performed with appropriate fetal shielding and limited late in gestation when the gravid uterus is pressing against the diaphragm. Computed tomography (CT) scans should be avoided during pregnancy because of the radiation exposure. Abdominal ultrasound for the evaluation of liver metastases is safe, but in pregnant women, significantly less sensitive than CT or MRI. MRI without gadolinium can be considered only if needed, especially in the first trimester, since there is a limited experience assessing safety during organogenesis [32]. Bone scans must not be used in pregnant patients for the evaluation of bone disease in the absence of signs or symptoms of bone abnormality. As an alternative, skeletal MRI may be considered (without contrast). Increases in tumor markers CA (cancer antigen ) 15.3 and CEA (Carcinoembryonic antigen) always give rise to the suspicion of metastasis [33]. Locally advanced-stage disease and/or suspicious symptoms should prompt a complete radiographic staging evaluation with modifications and shielding to protect the fetus. Since the therapeutic approach to patients with early or metastatic breast cancer is not usually changed during pregnancy (neither targeted nor hormonal treatment is considered), it is possible to safely leave staging of early breast cancer examinations after delivery, preferably using PET-CT or CT scans [34].

## 9. Hereditary Breast Cancer and PABC

Genetic predisposition to breast cancer is more frequent among pregnant women with cancer. The protective effect of multiparity and breastfeeding may be lost in women who inherit BRCA2 (but not BRCA1) mutations. *BRCA1* (Breast cancer antigen 1) or *BRCA2* (Breast cancer antigen 2) mutations confer the women with a 50–80% lifetime risk of breast cancer and 16–65% lifetime risk of ovarian cancer. These risks far exceed those of breast (13%) and ovarian (1.5%) cancer in the general population [35].

Most cases of breast cancer related to BRCA1 and BRCA2 are diagnosed in young women, and the probability of pregnancy in young women is high. At present, several other genes that increase the risk of breast cancer (*PALB2, CHECK2, CDH1*, etc.) are being identified in genetic screening panels. Genetic counseling is recommended for all patient with PABC [36]. Carriers of BRCA/2 not only have a higher risk of developing PABC but also have probably poorer outcomes with higher probability of developing distant metastases [37].

If a pregnant woman carries a BRCA1/2 mutation, this information may influence the decision on the type of surgery but does not allow the use of PARP (poly-ADP ribose polymerase) inhibitors in pregnancy in case of metastatic spread.

## 10. Monitoring of the Pregnancy

The pregnant woman with breast cancer requires careful and continuous monitoring of her pregnancy by her obstetrician and her oncologist. Confirmation of gestational age and expected date of delivery are important, as both are significant factors in treatment planning. For this reason, follow-up should take place at the center with experience in the care of patients with BCP and the gynecologist/obstetrician should be the part of the multidisciplinary team [38]. Breast-feeding should be discontinued immediately after delivery. Since, according to clinical studies, a properly selected cancer treatment does not compromise the cognitive function of the newborn as opposed to its immaturity, it is optimal to complete pregnancy until physiological delivery, if this is possible in terms of the severity of the disease course [39].

## 11. Prognosis

Based on smaller studies, maternal outcome may be worse in women with breast cancer diagnosed in pregnancy [40]. The largest cohort study in women treated for PABC, however, demonstrated similar disease-free survival and overall survival comparable to those of the general population [41].

In the registry study that compared over 300 women with breast cancer during pregnancy with almost 870 women who were not pregnant at the time of diagnosis, there was no significant difference in either progression-free survival (PFS, hazard ratio (HR) 1.34, 95% CI 0.93–1.91) or overall survival (OS, HR 1.19, 95% CI 0.73–1.93) [42]. In another smaller study that included 75 women who received standard chemotherapy during the second and third trimesters, women who were pregnant had a significantly improved five-year disease-free survival (72% vs. 57%) and OS (77% vs. 71%) [43].

A 2012 meta-analysis comprising over 3000 cases of gestational breast cancer and 37,100 controls found that gestational breast cancer was associated with a higher risk of death (HR 1.44, 95% CI 1.27–1.63), however, the association appeared to be limited primarily to women diagnosed in the postpartum period (HR 1.84, 95% CI 1.28–2.65) rather than during pregnancy (HR 1.29, 95% CI 0.72–2.24) [44].

## 12. Treatment of BCP

Pregnant women with breast cancer should be treated according to the guidelines for non-pregnant patients, with some modifications to protect the fetus (Table 3) [42,45].

### 12.1. Surgery

Either breast-conserving surgery or mastectomy are a reasonable option for the pregnant woman with breast cancer. A choice between them is guided by tumor characteristics and the result of the genetic test and patient preferences [49]. Women with breast cancer during pregnancy should undergo an axillary node evaluation. While axillary lymph node dissection is preferred, there are increasing data on the safety and efficacy of sentinel lymph node dissection [50].

The best cosmetic results and the least complications are achieved by surgery on a hormonally unstimulated breast preferably after childbirth after lactation arrest.

### 12.2. Radiotherapy

If the breast-conserving surgery is performed, the adjuvant radiotherapy (RT) should be postponed after delivery. The threshold for adverse radiation effects in fetuses is less than 100 mGy. Given the high dosage of fetal radiation, radiation therapy for breast cancer in pregnancy is still considered an absolute contraindication, although this may change in coming years with improving technologies [51].

As methods of stereotactic radiation and improved modalities of delivery are developed, radiation therapy may be an option for more women during pregnancy [46].

### 12.3. Systemic Antitumor Therapy


**Pharmacokinetics and Distribution of Drugs in Pregnancy**


Alterations in drug distribution are expected due to the physiologic changes that occur in pregnancy. Pregnancy leads to 40–60% increase in plasma volume even as early as 6 weeks after gestation. Increased fluid volume is associated with decreased plasma albumin, which may interfere with plasma concentration of some protein-bound drugs, e.g., taxanes, but this effect may be counterbalanced by high levels of estrogens, which increase other plasma proteins. Drug clearance by the kidney and liver increases, which may again reduce plasma levels of cytotoxic drugs. Diminished gastric motility may impact the absorption of orally administered drugs. “Third space” of the amniotic sac may play a role as well. The multidrug-resistance p-glycoprotein has been detected in fetal tissues and in the gravid endometrium and may offer some degree of protection to the fetus. However, currently it is not clear how these physiologic changes impact upon active drug concentrations and their resulting efficacy and toxicity. Moreover, pregnant women receive similar body surface-area based chemotherapy doses as non-pregnant women, which are adjusted according to continuing weight gains [52].
**Chemotherapy**

Patients indicated to chemotherapy during pregnancy may only start treatment after the first trimester. Data are available namely for anthracycline-based chemotherapy, often on an every-three-week schedule. Anthracyclines, more specifically doxorubicin, have not been found to significantly affect the cardiac function of children exposed in utero [53]. However, at least four cases of neonatal adverse cardiac effects have been reported after in utero exposure to anthracyclines, and there are several cases of in utero fetal death after exposure to idarubicin or epirubicin. Largely because of these reports, doxorubicin is preferred to idarubicin or epirubicin for the use in pregnancy [54]. Cyclophosphamide also has not been demonstrated to increase neonatal morbidity. In a prospective single-arm study, 87 pregnant breast cancer patients were treated with FAC (5-fluorouracil, adriamycine (doxorubicine), cyclophosphamide) in the adjuvant or neoadjuvant setting [55]. No stillbirths, miscarriages, or perinatal deaths occurred in the cohort of patients who received FAC chemotherapy during their second and/or third trimester. Most of the children did not have any significant neonatal complications. Three children were born with congenital abnormalities: one each with Down syndrome, ureteral reflux, or clubfoot. The rate of congenital abnormalities in the cohort was similar to the national average of 3%.

Taxanes, specifically paclitaxel, have not been found to be teratogenic when administered in the third trimester. Paclitaxel is preferred over docetaxel due to the better transplacental transfer of docetaxel. Taxanes were administered in the second and third trimesters in 38 patients and for the treatment of breast cancer in 27 patients. Despite the limitations and bias inherent in case reports, the use of taxanes appears feasible and safe during the second and third trimesters of pregnancy, with minimal maternal, fetal, or neonatal toxicity [56]. Although taxanes have promising treatment outcomes, we still have information about their safety only from case reports and small case series, and therefore, we must use them with caution [57]. Platinum derivatives may play a role in the treatment of triple negative breast cancer. They are highly protein bound, but the unbound fraction may cross the placenta. Carboplatin may be associated with the derangements of trophoblast invasion and disrupting placental development, which is not complete until 20 weeks of gestation. Although the data regarding the safety of platinum in pregnancy are limited, a systematic review of the use of carboplatin and cisplatin in pregnancy found that no malformation or toxicity was reported in seven carboplatin-exposed neonates [58]. Although only limited case reports are available, anthracycline chemotherapy administered on a dose-dense schedule (i.e., treatment every two weeks) does not appear to increase the risks of maternal or fetal complications compared with treatment administered every three weeks [59]. Chemotherapy should be avoided for three to four weeks before delivery whenever possible to avoid transient neonatal myelosuppression and potential complications, including sepsis and death. Weekly regimens with low hematotoxicity are an exception [60].
**Targeted Treatment**

The use of trastuzumab during pregnancy is relatively contraindicated. Exposure to trastuzumab during pregnancy can result in oligohydramnios, which in some cases may lead to pulmonary hypoplasia, skeletal abnormalities, and neonatal death. Women exposed to trastuzumab during pregnancy require ongoing monitoring of amniotic fluid volume, which is a marker of fetal renal status, throughout the pregnancy [61,62]. In a case report of maternal exposure to lapatinib for 11 weeks during the first and second trimester of pregnancy, there was an uneventful delivery of a healthy female infant, who was developmentally normal at 18 months of age [63].

However, until more information is available, we recommend against the use of lapatinib during pregnancy and lactation. There are currently no significant data on the safety of other anti-HER2 agents such as pertuzumab and ado-trastuzumab emtansine (TDM-1), and therefore, we do not recommend these agents until after delivery. However accidental short-term exposure to these agents during the first trimester does not appear to be associated with increased risk of fetal malformation, which is different compared to the risk from chemotherapy [64].

Currently we have not enough information on the safety of using bevacizumab, PARP inhibitors, and immunotherapy (PD-1 (Programmed death-1) and PDL-1 (Programmed death ligand-1) inhibitors) during pregnancy.
**Endocrine Treatment**

The use of selective estrogen receptor modulators (SERMs) such as tamoxifen during pregnancy should be generally avoided. They have been associated with vaginal bleeding, ambiguous genitalia, miscarriage, congenital malformations (spinal abnormalities, absent ears, craniofacial abnormalities, and cardiac malformation seen in Goldenhar’s syndrome), and fetal death [65]. Aromatase inhibitors (AIs) and luteinizing hormone-releasing hormone (LHRH) agonists are both contraindicated in pregnancy. AIs are not used in premenopausal women, but AIs combined with ovarian suppression by LHRH agonists may be used following term delivery.
**Supportive Care**

Antiemetics, including selective serotonin (5-HT) and neurokinin 1 (NK1) antagonists, are used to treat severe nausea and vomiting in pregnant women and are generally considered safe. However, long-term dexamethasone therapy should be avoided, if possible, because of potential maternal and fetal risks. Safe use of G-CSF (Granulocyte-colony stimulating factor) (and recombinant erythropoietin) in human pregnancy has been reported. Although there are no prospective trials evaluating the use of G-CSF or granulocyte-macrophage colony-stimulating factor (GM-CSF) in pregnant women, these agents are safe in the treatment of neonatal neutropenia and/or sepsis, but more caution is needed considering the very limited data. Hence, dose-dense chemotherapy is not the optimal strategy in pregnant patients [66].

### 12.4. Postponement of Treatment

If a malignant tumor is diagnosed in the first trimester, it is possible to terminate the pregnancy prematurely or postpone treatment until the second trimester. Delay can mean the risk of progression and generalization of the disease depending on the type of cancer and its staging at the time of diagnosis and may worsen prognosis (Table 4) [67]. If the patient has a lower-grade hormone-dependent cancer limited to the breast itself, the risk of delay is lower than in triple-negative cancer with nodal involvement. Delaying chemotherapy by 3–6 months may increase the risk of metastases by 5–10% [68].

### 12.5. The Course of Pregnancy, Fetal Monitoring, and Childbirth

Based on the available evidence, chemotherapy in BC patients may be safe during the second and third trimesters, with cessation of treatment three weeks prior to expected delivery. The most common complications of pregnancy associated with the application of chemotherapy are intrauterine growth retardation, prematurity, low birth weight, and bone marrow toxicity. Prematurity is generally associated with worse neonatal and long-term outcomes and, thus, should be avoided. Fetal condition can be well monitored by regular ultrasound biometrics and Doppler flowmetry. If premature birth is necessary, induction of fetal pulmonary maturity by corticoid administration is indicated. Most women expect vaginal delivery at term, but due to chemotherapy, delivery must be planned and induced, and immediately after delivery, lactation must be stopped.

## 13. Infant Outcome

Data suggest that early development among children born to women with cancer appears similar to that of children of the same gestational age, irrespective of in utero exposure to radiation or chemotherapy.

In a study of 129 children born to mothers diagnosed with cancer during pregnancy (over half of whom had breast cancer), cardiac, cognitive, and general development after a median of 22 months was equivalent with controls matched for gestational age [69]. In a subgroup analysis of children exposed to anticancer therapy in utero, similar outcomes were reported for the 96 children exposed to chemotherapy after the first trimester and the 11 children exposed to radiation compared with gestational-age-matched controls. There was a non-significant trend toward higher rates of small for gestational age at birth infants born to women with cancer (22% vs. 15%), particularly if exposed to chemotherapy or radiation. While the median gestational age of the children born to women with cancer was 36 weeks and, thus late preterm, it is unclear whether these children were born early because of early induction given their mothers’ diagnosis of cancer.

In the cohort study of 1170 pregnant women with all types of cancer treated at multiple institutions, 39% of whom had breast cancer, 88% of pregnancies resulted in live births [70]. Half of these deliveries were preterm, almost 90% of which were iatrogenic. These studies suggest that low neonatal complication rates are associated with in utero exposure to chemotherapy, but long-term data are limited. Moreover, studies may be limited by the fact that treatment providers may sometimes opt for early delivery induction, even when pregnancy does not affect treatment. One study reported 40% mortality among patients with advanced BCP who received chemotherapy when studied over a 13-year period (1991–2004) [71]. For women with breast cancer during pregnancy, the risk of cancer to the unborn is unknown, although there are no reported cases of childhood cancer arising in children exposed to chemotherapy of their mothers for breast cancer in utero.

## 14. Termination of Pregnancy

Early termination of pregnancy does not improve the outcome of BCP. In fact, some series suggest decreased survival in pregnant women who electively terminate their pregnancies compared with that in those who continue the pregnancy. However, these studies are retrospective case reviews and possible bias cannot be excluded; women with more advanced disease or poorer prognostic features possibly were more likely to be counseled to have an abortion [71]. The decision to terminate pregnancy for health reasons is difficult and should always be comprehensively considered in terms of the risk of fetal cancer treatment, the patient’s prognosis, and the impact of cancer therapy on the mother’s fertility. Although this situation is quite ambiguous, many physicians recommend to the patients with BCP to end pregnancy and so often deprive the patient of their only chance of having a child (Table 4).

## 15. Metastatic BCP

During pregnancy we can also diagnose patients with de novo metastatic breast cancer, and some patients with early breast cancer treated in a neo/adjuvant setting later metastasize. The main problem of the care of the metastatic breast cancer in pregnancy is limited treatment options with respect to the fetus. The main goal of therapy is to prolong the patient’s life, maintain its quality, not to damage the fetus, and for mother to spend as much time as possible with the child. This situation is extremely physically and psychologically demanding for the patient and affects the whole extended family [72].

## 16. Tailoring Treatment of Breast Cancer in Pregnancy

Personalized medicine has changed our approach from a “one size fits all” to the treatment of patients in a more individually tailored way. The goal of clinical research programs with a personalized approach to patients with breast cancer is to evaluate the unique code of RNA and DNA of cancer, enabling individualization of the treatment plan [73].

During pregnancy, tailoring to immunohistochemical markers such as hormone receptors, HER2 or PDL-1 expression, cannot be used at present, due to the risk of fetal harm. Genome testing and the use of next-generation sequencing (NGS) could, in the future, refine the prognosis of cancer and its sensitivity to chemotherapy, as the only acceptable systemic treatment in pregnancy.

From 2010 to 2020, 53 patients with BCP were treated at the Department of Oncology of the First Faculty of Medicine and the General Hospital in Prague. The number and proportion of patients has been influenced by the fact that in our comprehensive cancer center we have a program dedicated to young patients under 35 years of age and pregnant patients with breast cancer are referred to us from almost all over the Czech Republic (Table 5).

## 17. Conclusions

BCP is an example of cancer where individualization of the treatment approach could significantly improve the results of treatment and the hope of patients with concomitant breast cancer and pregnancy to prolong survival. The therapeutic plan must be adapted to the clinical parameters, the degree of pregnancy, the type and stage of the tumor, and the patient’s preference. The current options for a personalized treatment approach are not yet widely used in this subgroup of patients, although, in the future it would certainly be possible to focus molecular biology, NGS, and liquid biopsy methods to refine staging, estimate tumor chemosensitivity, and cancer prognosis to assess possible postponement of treatment to the postpartum period. Physicians treating patients with breast cancer in pregnancy have increased responsibility because they are trying to save two lives. While information and data on BCPs are increasing, it is necessary to centralize the treatment of BCP in the hands of experienced oncologists and obstetricians with praxis in this type of high-risk pregnancy and personalized access to each pregnant patient.

## Figures and Tables

**Table 1 jpm-11-00018-t001:** Tailoring treatment according to the type of breast cancer.

Tumor Subtype	Luminal A	Luminal B	HER2+	Triple Negative
Preferred approach	Surgery, postponement of hormone therapy, and radiotherapy after delivery	Surgery, adjuvant/ neoadjuvant chemotherapy, depending on the stage, postponement of hormone therapy, and radiotherapy after delivery	Surgery, adjuvant/ neoadjuvant chemotherapy depending on the stage, postponement of anti-HER2 treatment, and radiotherapy after delivery	Surgery, adjuvant/ neoadjuvant chemotherapy depending on the stage, postponement of radiotherapy after delivery

HER2, human epidermal growth factor receptor 2.

**Table 2 jpm-11-00018-t002:** Tailoring treatment according to the stage of breast cancer.

Stage	Local	Local Advanced	Metastatic
Treatment approach	Surgery with subsequent adjuvant chemotherapy, hormone therapy, targeted therapy, and radiotherapy must be postponed after delivery	Neoadjuvant chemotherapy, subsequent surgery usually after delivery, hormone therapy, targeted therapy, and radiotherapy must be postponed after delivery	Palliative chemotherapy in pregnancy, targeted treatment, hormone therapy, must be postponed after delivery

**Table 3 jpm-11-00018-t003:** Personalization of breast cancer treatment in pregnancy with regard to its stage (adapted according to [46,47,48]).

Stage	Early First Trimester Conception—4 Weeks	First Trimester 4—14 Weeks	Second Trimester 14 Weeks—28 Weeks	Third Trimester 28 Weeks—Delivery
Surgery	1–2% increased risk of miscarriage	1–2% increased risk of miscarriage	Premature delivery	Premature delivery
Radiotherapy	All or none	Gross malformation, microcephaly, mental retardation	Mental and growth retardation, cataracts, microcephaly, sterility, secondary malignancies	Growth retardation, sterility, cataracts, secondary malignancies
Gamma Knife stereotactic radiosurgery (GKSRS)	Lack of data	Lack of data	Probably safe by a conservative treatment of patients with multiple brain metastases	Probably safe by a conservative treatment of patients with multiple brain metastases
Chemotherapy	All or none	High risk of severe fetal malformation. Increased risk of miscarriage	Growth restriction, low birth weight, preterm labor, myelosuppression, need for neonatal intensive care unit admission	Growth restriction low birth weight, preterm labor, myelosuppression, need for neonatal intensive care unit admission
Anti-HER2	Fetus unaffected in review of limited case reports	Fetus unaffected in review of limited case reports	Oligohydramnios/anhydramnios	Oligohydramnios/anhydramnios
Hormonal therapy	Possible increased risk of miscarriage	Facial malformations, ambiguous genitalia, possible increased risk of miscarriage, some cases with no adverse effects observed, data limited to animal studies and case reports	Insufficient data	Insufficient data
Immunotherapy	Increased risk of miscarriage	Increased risk of miscarriage	Increased risk of stillbirth, premature delivery, infant mortality	Increased risk of stillbirth, premature delivery, infant mortality
Anti-VEGF/VEGFR (Vascular endothelial growth factor/Vascular endothelial growth factor receptor)	All or none	Increased risk of miscarriage, skeletal malformations, abnormal vascular development of the skin, pancreas, kidney, and lung	Intrauterine growth restriction, preeclampsia, hypertension	Intrauterine growth restriction, preeclampsia, hypertension
PARP inhibitors	Lack of data in pregnant women	Potential to cause embryo-fetal harm, but lack of data	Potential to cause embryo-fetal harm, but lack of data	Potential to cause embryo-fetal harm, but lack of data

**Table 4 jpm-11-00018-t004:** Personalization according to patient preference.

Patient Preference	Request	A Possible Solution
Staging	Avoid all imaging methods with radiation	Tumor markers, abdominal ultrasonography, MRI without contrast, until after delivery complete staging using PET-CT (Positron emission tomography—computed tomography) or CT(Computed tomography)
Termination of pregnancy	To prioritize the life of the mother over the life of the child	Does not bring any benefits in terms of overall survival, subsequent pregnancy is possible but uncertain, interruption must be considered in the first trimester of pregnancy, if the initiation of anticancer treatment cannot be delayed
Anticancer treatment in pregnancy	Avoid anticancer treatment during pregnancy due to concerns about the baby	Treatment can be delayed with varying degrees of risk of progression and generalization depending on the type of cancer, the patient must be informed of the risks of delay and the fact that properly timed surgery and chemotherapy do not pose a serious risk to the fetus
Spontaneous vaginal delivery	Avoid a planned cesarean delivery	The reason for the planned delivery is the risk of severe neonatal life-threatening neutropenia of the fetus after chemotherapy, in pregnant women treated with a weekly chemotherapy regimen (e.g., taxol), it is possible to consider spontaneous delivery

**Table 5 jpm-11-00018-t005:** Patients with breast cancer occurring during pregnancy (BCP) were treated at the Department of Oncology of the First Faculty of Medicine and the General Hospital in Prague (2010–2020).

N	Termination Pregnancy	BRCA1+/BRCA2+	Local Recurrence	De Novo Metastatic	Systemic Recurrence	Median Age
53	3	4/2	1	7	14	31 years

## Data Availability

Not applicable.

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
