# Peer review of "Is It Possible to Personalize the Diagnosis and Treatment of Breast Cancer during Pregnancy?"

_jpm, 2020, doi:10.3390/jpm11010018_

Round 1

Reviewer 1 Report

Dear authors, 

you propose an interesting review about breast cancer during pregnancy, which is a real and complex issue of the health system. 

But I want to highlight that some parts of the manuscript have deficiencies in references: 

ex. from line 125 to 145 - just reference [20] ; from line 147 - 172 - just 3 [ 21,22,23]. Long paragraphs with one or 2 references should not appear in a review. We can correct it. 

Overall, the paper is clear and well written.

Author Response

I added several paragraphs dedicated exclusively to the personalized treatment with several further references (so the references need to renumbered). The text has been also corrected by native speaker, so I believe that the quality of language improved.

Reviewer 2 Report

This current article depicts a comprehensive overview of pregnancy associate breast cancer including diagnosis, stages, available therapies and effect on the fetus. Authors also went over why precision medicine approach is ideal in this disease. Overall the article covers this disease in great extent but the precision medicine sections are not elaborated very well. Please see my comments below.

  1. It will be great if the authors can add a section introducing the precision medicine in the 'Introduction' section along with PABC.
  2. Please elaborate on advancement of precision medicine in cancer in general with primary focus on breast cancer. Authors can further utilize this section to rationalize the current advancement and future potential involvement of precision medicine in pregnancy associated breast cancer.
  3. It will be great if the authors can edit the manuscript to incur a better writing style and sentence structures overall to convey the message more efficiently.

Author Response

(The authors gave the same response as above.)
